# Aviation Mutagenesis Alters the Content of Volatile Compounds in Dahongpao (*Camellia sinensis*) Leaves and Improves Tea Quality

**DOI:** 10.3390/foods13060946

**Published:** 2024-03-20

**Authors:** Jianghua Ye, Qi Zhang, Pengyuan Cheng, Yuhua Wang, Jishuang Zou, Shaoxiong Lin, Mingzhe Li, Miao Jia, Yiling Chen, Xiaoli Jia, Haibin Wang

**Affiliations:** 1College of Tea and Food, Wuyi University, Nanping 354300, China; jhye1998@126.com (J.Y.); zhangqi1113@126.com (Q.Z.);; 2College of Life Science, Longyan University, Longyan 364012, China; 3College of JunCao Science and Ecology, Fujian Agriculture and Forestry University, Fuzhou 350002, China

**Keywords:** aviation mutagenesis, *Camellia sinensis*, volatile compounds, odor characteristics, quality

## Abstract

Aviation mutagenesis is a fast and efficient breeding method. In this study, we analyzed the effect of aviation mutagenesis on volatile compounds and odor characteristics in Dahongpao fresh leaves and gross tea for the first time. The results showed that aviation mutagenesis significantly increased the total volatile compounds of Dahongpao fresh leaves and gross tea. Aviation mutagenesis most critically significantly increased the content of beta-myrcene in Dahongpao fresh leaves, prompting its conversion to beta-pinene, cubebol, beta-phellandrene, zingiberene, (*Z*,*Z*)-3,6-nonadienal, and 6-pentyloxan-2-one after processing, which in turn enhanced the fruity, green, spicy, and woody odor characteristics of the gross tea. This study provided a reference for further exploration of aviation mutagenic breeding of *Camellia sinensis*.

## 1. Introduction

Wuyi Mountain, Fujian, China (27°32′36″~27°55′15″ N, 117°24′12″~118°02′50″ E) is an important tea-producing area in China, the birthplace of oolong tea. Wuyi Mountain has uniform temperature in all seasons, with an annual average temperature of 12~13 °C, which is mild and humid, and its geology belongs to the typical Danxia landform; the special geographic environment creates a special type of tea—Wuyi Rock tea. Second, Wuyi Mountain is a dual world of cultural and natural heritage, a global biodiversity conservation area, and a world biosphere reserve. In order to protect Wuyi Mountain’s special natural resources and maintain the ecological environment, Wuyi Mountain prohibits the cultivation of additional tea plantations, which can only be managed or replanted on the sites of the original tea plantations [1,2]. From 2012 to 2022, the area of tea plantations in Wuyi Mountain basically stabilized at 10,000 ha, and the gross tea production was about 3000 t [3]. With the promotion of the Wuyi Rock tea brand and the growing industry, sales of Wuyi Rock tea have been increasing, and existing tea plantations can no longer meet market demand [4] Therefore, based on existing tea plantations, tea farmers have only been able to improve tea yield and quality by continuously optimizing the management of tea plantations; however, they have had little success [5].

In recent years, many scholars have carried out a lot of research on how to optimize management measures for tea plantations and thus improve the yield and quality of tea. For example, fertilization, irrigation, pest control, and pruning, while effective in improving tea yield and quality, still fail to meet market demand [6,7,8,9]. Therefore, some scholars believe that the key to maintaining the sustainable development of the Wuyi Rock tea industry is to optimize the selection of high-yield and high-quality tea germplasm resources [10,11]. However, under natural conditions, artificial cultivation of high-quality tea germplasm resources often requires long cycles, which is still a drawback to the rapid promotion of the Wuyi Rock tea industry [12]. Aviation mutagenesis is a novel breeding technology in which plant seeds undergo significant changes in their growth, development, and physiological characteristics after being subjected to magnetic fields, gravity, and radiation in space, which is conducive to the rapid and efficient screening of high-quality germplasm resources [13,14]. For a long time, aviation breeding has been greatly emphasized in the world and has successfully obtained a number of superior varieties with strong resistance, high yield and high quality [15]. China has always attached great importance to aviation breeding, and as of 2021, about 200 new varieties of space-radiated plant have been developed, and these varieties have played a great role in promoting the development of Chinese agriculture [16]. In 2003, China’s “Shenzhou V” spacecraft took tea tree seeds into space for mutagenesis for the first time. As of November 2022, 12 batches of tea trees of different varieties have gone into space with the Shenzhou series of spacecraft for space mutagenesis. However, no studies have been reported to date on whether the yield and quality of tea trees change after aviation mutagenesis. In November 2011, tea tree seeds of Wuyi Rock tea with the Shenzhou VIII spacecraft were sent into space for space mutagenesis; so far, this is the only batch of aviation mutagenic Wuyi Rock tea. After the above aviation mutagenic tea tree seeds landed on the ground, they were bred at the same time as the ground control tea seeds in 2012. After 11 years, the tea trees are now in their normal growth and harvesting period. No report has been made on whether the quality of Wuyi Rock tea changes after aviation mutagenesis. This study analyzes in depth whether aviation mutagenesis will have an effect on the formation of tea tree quality, which is of great significance for the development of aviation breeding of tea trees and the sustainable development of the tea industry.

Aroma is an important index for assessing tea quality, and the level of tea aroma quality is closely related to the content of volatile compounds in tea [17]. There are many types of volatile compounds in tea, and different volatile compounds present different odor characteristics, while the intensity of odor characteristics of tea can be determined by judging the content of volatile compounds, which in turn evaluates the aroma intensity of tea [18]. Therefore, the determination of volatile compounds in tea is commonly used in the evaluation of tea aroma quality [19]. Aviation mutagenesis is one of the most important techniques in plant breeding, and significant changes in plant growth, metabolism, and chemical composition can be achieved through aviation mutagenesis [20]. For example, the composition of volatile compounds in *Andrographis paniculata* was essentially similar to that of control after aviation mutagenesis, but the content changed significantly, especially aldehydes [15]. Accordingly, this study was conducted to analyze the effects of aviation mutagenesis on the volatile compound content and odor characteristics in fresh leaves and gross tea of Dahongpao, using Wuyi Rock tea Dahongpao tea trees that were subjected to aviation mutagenesis and Dahongpao tea trees that were left on the ground as materials. Meanwhile, characteristic volatile compounds and their odor characteristics that undergo significant changes were screened and obtained in Dahongpao fresh leaves and gross tea after aviation mutagenesis. On this basis, the transformation mechanism of characteristic volatile compounds in fresh leaves and gross tea was further analyzed, with a view to revealing the mechanism of aviation mutagenesis in the formation of Dahongpao aroma quality and providing a reference for the development of aviation breeding of tea trees.

## 2. Materials and Methods

### 2.1. Materials

The experimental tea plantation in this study was located at the Tea Tree (*Camellia sinensis*) aviation Breeding Experimental Base of Xianming Rock Tea Factory, Wuyishan City, Fujian Province, China (117°59′47.7″ E, 27°44′8.4″ N), and the tea tree variety was Dahongpao (*Camellia sinensis*), with an age of 11 years (Appendix A). The process of aviation mutagenesis and cultivation of Dahongpao tea tree seeds was as follows: at 5:58 p.m. on 1 November 2011, Dahongpao tea tree seeds were launched with the unmanned spacecraft Shenzhou VIII, and two days later, Shenzhou VIII docked with the target vehicle Tiangong I in a space rendezvous. After 12 days of operation, the Shenzhou VIII spacecraft and the Tiangong I target vehicle were successfully separated at 18:30 on 16 November 2011. The capsule returned to Earth at 19:32 on 17 November 2011, with a total flight time of 16 days, 13 h and 34 min. The aviation mutagenic seeds and control seeds were germinated and cultivated in the same plot in April 2012 and were managed in the same manner during cultivation.

Fresh leaves of Dahongpao tea tree with aviation mutagenesis (TM) and without aviation mutagenesis (CK) were collected in May 2023, and headspace solid-phase microextraction (HS-SPME) was used to extract and enrich volatile constituents in fresh leaves of Dahongpao tea tree; volatile constituents were identified by gas chromatography–mass spectrometry (GC-MS), with three independent replicates for each sample. Fresh leaves of the tea tree were collected (one bud and three leaves, as per the traditional tea picking method) with three independent replicates per treatment and 50 kg per replicate. Of these, 2 kg of fresh leaves were taken from each replicate after thorough mixing for the determination of volatile compounds in the leaves, and the remaining 48 kg were used for the primary processing of tea leaves. Fresh leaves of tea trees were processed according to the Wuyi Rock tea standard primary processing method, which mainly includes five stages: fresh leaf picking, withering, fermentation, kneading, and drying, with three independent replicates for each sample [21]. The specific process flow and method are shown in Appendix A. After completion of primary processing, 2 kg each of FTM and FCK of gross tea after primary processing of TM and CK were collected, respectively, with three independent replicates for each treatment. The collected gross tea samples were used to identify volatile compounds by HS-SPME combined with the GC-MS technique.

### 2.2. Volatile Metabolome Analysis

Fresh leaves of the tea tree and gross tea were ground into powder in liquid nitrogen. To prevent any enzyme reactions, 500 mg of the powder was quickly transferred immediately to a 20 mL head-space vial (Agilent, Palo Alto, CA, USA), which was filled with a NaCl saturated solution. The vials were securely sealed using crimp-top caps with TFE-silicone headspace septa (provided by Agilent). Upon performing solid-phase microextraction (SPME) analysis, each vial was initially heated to 60 °C for 5 min. A 120 µm DVB/CWR/PDMS fiber (Agilent) was then inserted into the headspace of the sample at this same temperature, allowing for a 15 min exposure.

Following the sampling process, the fiber coating’s volatile organic compounds were desorbed at the injection port of the GC apparatus (Model 7890B, Agilent) for 5 min at a temperature of 250 °C in the splitless mode. The detection and quantification of VOCs was performed using an Agilent Model 7890B GC coupled with a 7000D mass spectrometer (Agilent). The GC system was equipped with a 30 m × 0.25 mm × 0.25 μm DB-5MS (5% phenyl-polymethylsiloxane) capillary column. Helium gas was used as a carrier gas at a linear velocity of 1.2 mL/min. The injector temperature was maintained at 250 °C and the detector at 280 °C. The oven temperature was programmed to increase gradually from 40 °C (3.5 min) to 100 °C at a rate of 10 °C/min. This was followed by a ramp up to 180 °C at a rate of 7 °C/min. After this, the oven temperature was increased to 280 °C, holding for 5 min before the next measurement cycle. Mass spectra were collected with electron impact (EI) ionization mode at 70 eV. During this experiment, the quadrupole mass detector, ion source, and transfer line temperatures were adjusted to 150, 230 and 280 °C, respectively. The mass spectrometer was set to SIM mode for the identification and quantification of the analytes. The qualitative and quantified methods of volatile metabolites were as follows. One quantitative ion and two to three qualitative ions were selected for each compound. All the ions to be detected in each group were detected separately by time period according to the order of peaks. If the retention time of the detected ions was consistent with the standard reference value, and the selected ions all appeared in the mass spectra of the samples after deduction of the background, the substance could be judged as the substance. The database used for the comparison was the NIST20 mass spectrometry database. It was also necessary to select quantitative ions for integration and correction work to improve the accuracy of quantification [22].

### 2.3. Statistical Analysis

Excel 2017 software was used to pre-process the raw data, including the classification, number calculation, and content statistics of volatile compounds. Rstudio software (version 4.2.3) was used to produce box plots (gghalves 0.1.4), principal component plots (ggbiplot 0.55), heat maps (pheatmap 1.0.12), and bubble feature maps (ggplot2 3.4.4); it was also used to construct a partial least squares-discriminate analysis model (OPLS-DA, R library was ropls and mixOmics) and odor wheel (ComplexHeatmap version 2.16.0, circlize version 0.4.15, vegan version 2.6.4, RColorBrewer version 1.1.3). The TOPSIS analysis was performed on the SPSSAU online platform (https://spssau.com/ accessed on 10 November 2023) [23].

## 3. Results and Discussion

### 3.1. Analysis of the Number and Classification of Volatile Compounds in Dahongpao Fresh Leaves and Gross Tea

In this study, GC-MS was used to analyze the effect of aviation mutagenesis on volatile compounds in fresh leaves and gross tea from Dahongpao. Analysis of volatile compounds in Dahongpao fresh leaves showed (Figure 1A) that 230 and 218 volatile compounds were detected in aviation mutagenic (TM) and control (CK) Dahongpao fresh leaves, respectively. There were 14 volatile compounds specific to CK, 26 volatile compounds specific to TM, and 204 identical volatile compounds. Classification analysis of volatile compounds detected in CK showed (Figure 1B) that 218 volatile compounds could be classified into 14 categories, of which the top 5 most abundant compounds were terpenoids (19.72%), heterocyclic compounds (16.97%), ester (14.22%), heterocyclic compounds (13.76%), and alcohol (9.17%), accounting for 73.84% of total volatile compounds. Classification analysis of the volatile compounds detected in TM showed (Figure 1C) that 230 volatile compounds could be classified into 15 categories, of which the top 5 compounds with the highest number were terpenoids (18.70%), heterocyclic compounds (16.97%), hydrocarbons (13.48%), ester (13.04%), and alcohol (10.43%), accounting for 72.62% of total volatile compounds.

In addition, this study further analyzed the changes in the number and categories of volatile compounds in gross tea after CK and TM were processed according to the Wuyi Rock tea production process. The analysis of volatile compounds in Dahongpao gross tea showed (Figure 1D) that 369 and 360 volatile compounds were detected in gross tea made from fresh leaves of aviation mutagenic (FTM) and control (FCK) Dahongpao, respectively. Second, there were 17 volatile compounds specific to FCK, 23 volatile compounds specific to FTM, and 343 volatile compounds that were the same in both. Classification analysis of the volatile compounds detected in FCK showed (Figure 1E) that the 360 volatile compounds could be classified into 16 categories, of which the top 5 categories with the highest number of compounds were terpenoids (25.56%), ester (16.94%), heterocyclic compounds (13.61%), ketone (9.17%), and hydrocarbons (8.89%), accounting for 74.17% of total volatile compounds. Classification analysis of the volatile compounds detected in the FTM showed (Figure 1F) that 369 volatile compounds could be classified into 16 categories, of which, the top 5 compounds with the highest number were terpenoids (25.68%), ester (15.57%), heterocyclic compounds (14.21%), hydrocarbons (9.56%), and ketone (9.02%), accounting for 74.04% of total volatile compounds.

It can be seen that after aviation mutagenesis, the number and categories of volatile compounds in fresh leaves and gross tea of Dahongpao were more similar to those of the control, and aviation mutagenesis had a smaller effect on the composition of volatile compounds in the fresh leaves and gross tea of the tea tree (*Camellia sinensis*).

### 3.2. Analysis of Volatile Compounds in Dahongpao Fresh Leaves and Gross Tea

Plants are affected by the take-off and landing of spacecraft during aviation mutagenesis and experience the strong effects of radiation and magnetic fields in space, which may alter the normal growth and development of plants and their metabolism and thus affect the content of different compounds in plant tissues [20]. It has been reported that *Prunella vulgaris* seeds carried by the Shenzhou VIII spacecraft which underwent aviation mutagenesis had a significantly higher rosmarinic acid content in the plant after planting than the control [24]. Secondly, it has also been found that glycyrrhizic acid and liquiritigenin contents were significantly elevated in plants of *Glycyrrhiza glabra* seeds after aviation mutagenesis compared to the control [25]. Based on the previous analysis, this study further analyzed the effect of aviation mutagenesis on the volatile compound content in fresh leaves and gross tea of Dahongpao. Analysis of the total volatile compound content of Dahongpao fresh leaves showed (Figure 2A) that the volatile compound content of TM was significantly higher (*p* < 0.05) than that of CK. PCA analysis of the relative content of volatile compounds detected in fresh leaves revealed (Figure 2B) that there was a significant difference in the content of different volatile compounds in CK and TM, and the two principle components could effectively differentiate between CK and TM with an overall contribution of 81.06%. After the classification of volatile compounds, the results of their content analysis showed (Figure 2C) that there were 13 categories of volatile compounds in TM with significantly higher content than CK, which were acid, aldehyde, amine, aromatics, ether, heterocyclic compounds, hydrocarbons, ketone, nitrogen compounds, phenol, sulfur compounds, terpenoids, and others. Moreover, the above 13 categories of volatile compounds were significantly associated with TM (Figure 2D). In contrast, the content of two categories of volatile compounds, such as alcohol and ester, was significantly lower in TM than in CK, and these two categories of volatile compounds were significantly correlated with CK (Figure 2C).

Further analysis of the effect of aviation mutagenesis on volatile compound content in Dahongpao gross tea showed (Figure 2E) that the total volatile compound content was significantly higher in FTM than in FCK (*p* < 0.05). PCA analysis with the relative content of volatile compounds detected in the gross tea revealed (Figure 2F) that there were significant differences in the content of different volatile compounds in FCK and FTM, and the two principle components could effectively distinguish FCK from FTM with an overall contribution of 84.39%. After the volatile compounds were categorized, the results of their content analysis (Figure 2G) showed that there were 13 categories of volatile compounds in FTM with significantly higher content than FCK, namely acid, aldehyde, amine, aromatics, ester, ether, halogenated hydrocarbons, heterocyclic compounds, hydrocarbons, ketone, nitrogen compounds, phenol, and terpenoids. Moreover, the above 13 categories of volatile compounds were significantly associated with FTM (Figure 2H). In contrast, the content of two categories of volatile compounds, such as alcohol and sulfur compounds, was significantly lower in FTM than in FCK, and these two categories of volatile compounds were significantly correlated with FCK (Figure 2H). There was no significant difference in the content of other volatile compounds between FTM and FCK (Figure 2H). It can be seen that aviation mutagenesis significantly altered volatile compound content in the fresh leaves and gross tea of Dahongpao, which in turn improved the aroma quality of Dahongpao tea.

### 3.3. Screening and Content Analysis of Characteristic Volatile Compounds in Dahongpao Fresh Leaves and Gross Tea

Based on the previous analysis, this study further screened the characteristic volatile compounds that changed significantly in Dahongpao fresh leaves and gross tea after aviation mutagenesis. A volcano plot analysis of volatile compound content in Dahongpao fresh leaves showed (Figure 3A) that 70 volatile compounds were significantly increased, 25 were significantly decreased, and 149 were not significantly different in TM compared to CK. The OPLS-DA model of CK and TM was further constructed to screen for key volatile compounds. The goodness-of-fit analysis of the OPLS-DA model showed (Figure 3B) that after 200 stochastic simulations of the constructed model, the model had R^2^Y = 1 and Q^2^ = 0.992, both at the *p* < 0.005 level. This result indicated that the OPLS-DA model constructed in this study could effectively distinguish between CK and TM, and the model fit met the requirements and could be used for further analysis. Analysis of the OPLS-DA model score plot for CK versus TM showed (Figure 3C) that the intra-group difference between CK and TM was within 2.99%, while the inter-group difference was 96.00%. This result showed that the reproducibility of three independent replicates of CK and TM in this study was good, and there was a significant difference in volatile compound content between CK and TM. On this basis, further analysis of the S-Plot plot of the OPLS-DA model of CK versus TM revealed (Figure 3D) that 47 volatile compounds (VIP > 1) were critically different in CK versus TM, of which 24 volatile compounds were significantly elevated in TM and 23 were significantly decreased in TM compared to CK. The classification analysis of the key differential volatile compounds showed (Figure 3E) that the 47 volatile compounds could be classified into 12 categories, of which, compared with CK, there was a significant increase in the content of 9 volatile compounds in TM, namely amine, aromatics, phenol, aldehyde, terpenoids, ketone heterocyclic compounds, and acid and sulfur compounds, whereas three categories of volatile compounds showed a significant decrease in the content, namely, alcohol, hydrocarbons and ester.

TOPSIS was further used to analyze the impact weights of the 12 categories of volatile compounds in distinguishing CK from TM, and the results showed (Figure 3F) that four categories of volatile compounds had impact weights greater than 10%, namely alcohol (100%), ester (62.04%), terpenoids (36.83%), and heterocyclic compounds (11.91%). A bubble characteristic plot with the above four categories of volatile compounds was performed to screen for characteristic volatile compounds that distinguish CK from TM. It was found (Figure 3G) that the characteristic volatile compounds distinguishing CK from TM were mainly trans-3-hexenyl acetate, beta-myrcene, 3-hydroxy-4-methyl-5-ethyl-2-furanone, (*Z*)-3-hexenyl butyrate, 2-p-tolylethanal, and 2-methyl-benzaldehyde. Analysis of the content of characteristic volatile compounds showed (Figure 3H) that beta-myrcene, 3-hydroxy-4-methyl-5-ethyl-2-furanone, 2-p-tolylethanal, 2-methyl-benzaldehyde were significantly higher in TM than in CK, whereas trans-3-hexenyl acetate and (*Z*)-3-hexenyl butyrate were significantly lower than in CK. It can be seen that the content of volatile compounds, especially characteristic volatile compounds, in the fresh leaves of Dahongpao changed significantly after aviation mutagenesis, and this change may affect the transformation of substances and the formation of aroma in the subsequent post-processing process of tea.

Accordingly, the volatile compound content in the gross tea of Dahongpao changed significantly after processing was analyzed in this study. Volcano plot analysis of volatile compound content in Dahongpao gross tea showed (Figure 4A) that there were 79 volatile compounds with a significant increase, 25 with a significant decrease, and 279 with no significant difference in FTM compared to FCK. The OPLS-DA model of FCK and FTM was further constructed to screen for key volatile compounds. The goodness-of-fit analysis of the OPLS-DA model showed (Figure 4B) that after 200 stochastic simulations of the constructed model, the model had R^2^Y = 0.999 and Q^2^ = 0.995, both at the *p* < 0.005 level. This result showed that the OPLS-DA model constructed in this study could effectively distinguish FCK from FTM, and the model fit met the requirements and could be used for further analysis. Analysis of the OPLS-DA model score plot for FCK versus FTM showed (Figure 4C) that the intra-group difference between FCK and FTM for the three independent replicates was within 0.96%, while the inter-group difference was 96.30%. This result showed that the reproducibility of three independent replicates of FCK and FTM in this study was better and there was a significant difference in volatile compound content between the two. On this basis, further analysis of the S-Plot plots of the OPLS-DA model of FCK versus FTM revealed (Figure 4D) that there were 77 volatile compounds with key differences in FCK versus FTM, of which 57 volatile compounds were significantly elevated and 20 significantly decreased in FTM compared to FCK. The classification analysis of key differential volatile compounds showed (Figure 4E) that 77 volatile compounds could be classified into 12 categories, of which 10 categories of volatile compounds were significantly increased in the FTM compared to the FCK, namely amine, aromatics, aldehyde, terpenoids, hydrocarbons, ketone, heterocyclic compound, ester, nitrogen compounds, and ether, while the content of two categories of volatile compounds, namely alcohol and phenol, decreased significantly.

TOPSIS was further used to analyze the impact weights of the 12 categories of volatile compounds in distinguishing FCK from FTM, and the results showed (Figure 4F) that the 6 categories of volatile compounds with impact weights greater than 10% were ester (96.82%), terpenoids (73.66%), heterocyclic compounds (66.40%), hydrocarbons (22.64%), aldehyde (17.55%), and ketone (11.55%). A bubble characteristic map analysis with six categories of volatile compounds revealed (Figure 4G) that the characteristic volatile compounds distinguishing FCK from FTM were mainly 6-pentyloxan-2-one, beta-pinene, cubebol, (*Z*,*Z*)-3,6-nonadienal, zingiberene, beta-phellandrene, and ethyl cinnamate. Analysis of the content of characteristic volatile compounds showed (Figure 3H) that FTM had a significantly higher content of 6-pentyloxan-2-one, beta-pinene, cubebol, (*Z*,*Z*)-3,6-nonadienal, zingiberene, and beta-phellandrene than FCK, whereas ethyl cinnamate was significantly lower than FCK. It can be seen that changes in volatile compound content in Dahongpao fresh leaves after aviation mutagenesis affected the volatile compound content in gross tea, especially the characteristic volatile compound content.

### 3.4. Odor Characteristic and Transformation Analysis of Characteristic Volatile Compounds in Dahongpao Fresh Leaf and Gross Tea

Changes in the content of volatile compounds can directly affect the type and intensity of aroma in tea [26]. It has been reported that beta-myrcene, 2-p-tolylethanal and 2-methyl-benzaldehyde are important compounds that make up the aroma of tea, and their main odor characteristic is floral [27]. The main odor characteristic of (*E*)-3-hexenyl acetate, 3-hydroxy-4-methyl-5-ethyl-2-furanone and (*Z*)-3-hexenyl butyrate is fruity [28,29]. In this study, odor wheel analysis of odor characteristics of characteristic volatile compounds in fresh leaves of Dahongpao revealed (Figure 5A) that the main odor characteristics presented by characteristic volatile compounds in CK and TM were floral and fruity, where the intensity of the floral odor characteristic in TM was significantly higher than that in CK, while the fruity odor characteristic was slightly lower than that in CK. In addition, this study identified seven characteristic compounds with significant differences in Dahongpao gross tea after aviation mutagenesis, namely ethyl cinnamate, 6-pentyloxan-2-one, (*Z*,*Z*)-3,6-nonadienal, beta-phellandrene, cubebol, zingiberene, and beta-pinene. The main characteristic odor of ethyl cinnamate is reported to be floral [30], that of 6-pentyloxan-2-one is fruity [31], that of (*Z*,*Z*)-3,6-nonadienal and beta-phellandrene is green [32,33], that of cubebol and zingiberene is spicy [34], and that of beta-pinene is woody [35]. In this study, the odor wheel analysis of odor characteristics of characteristic volatile compounds in Dahongpao gross tea revealed (Figure 5B) that the main odor characteristics presented by the characteristic volatile compounds were floral, fruity, green, spicy, and woody, where the odor intensity of fruity, green, spicy, and woody notes in FTM was significantly higher than that in FCK; the intensity of the floral odor was significantly lower than that in FCK. It can be seen that the intensity of the odor characteristics of Dahongpao fresh leaves and gross tea changed significantly after aviation mutagenesis. Aviation mutagenesis altered the content of characteristic volatile compounds in fresh leaves and gross tea of Dahongpao, which in turn affected the intensity of the tea’s odor characteristics.

Based on the previous analysis, this study further analyzed the transformation of volatile compounds in Dahongpao fresh leaves and gross tea. In this study, we constructed a transformational relationship of characteristic volatile compounds in Dahongpao fresh leaves and gross tea and found that (Figure 5C) beta-myrcene played an important role in the transformation of volatile compounds and was a precursor for the transformation or synthesis of other characteristic volatile compounds; we also found that this compound was mainly derived from fresh leaves. In Dahongpao fresh leaves, beta-myrcene content in TM was significantly higher than that in CK. Further analysis revealed that beta-myrcene could achieve the transformation of characteristic volatile compounds through three pathways. In the first pathway, beta-myrcene could be synthesized into beta-pinene; beta-pinene could be used to synthesize cubebol after carbon chain rearrangement or could be transformed by isomerization into beta-phellandrene; beta-phellandrene could be used for the synthesis of zingiberene or oxidation and ring-opening to (*Z*,*Z*)-3,6-nonadienal; (*Z*,*Z*)-3,6-nonadienal could be transformed by esterification into a ring to form 6-pentyloxan-2-one. In the second pathway, beta-myrcene could be synthesized into (*E*)-3-hexenyl acetate by oxidative esterification, into (*E*)-3-hexenyl acetate through hydrolytic esterification to synthesize (*Z*)-3-hexenyl butyrate, and into (*Z*)-3-hexenyl butyrate through hydrolytic ring opening and oxidative ring formation to synthesize 3-hydroxy-4-methyl-5-ethyl-2-furanone. Moreover, 3 -hydroxy-4-methyl-5-ethyl-2-furanone was synthesized by hydrolytic reduction of (*Z Z*)-3,6-nonadienal, and (*Z*,*Z*)-3,6-nonadienal could be transformed by esterification into a ring to form 6-pentyloxan-2-one. In the third pathway, beta-myrcene could be oxidized and cyclized to 2-p-tolylethanal; 2-p-tolylethanal could be synthesized to ethyl cinnamate by esterification or by oxidative rearrangement to form 2-methyl-benzaldehyde, and 2-methyl-benzaldehyde could be transformed to ethyl cinnamate by esterification.

From the first pathway of beta-myrcene transformation, there were six characteristic volatile compounds synthesized by its transformation, namely beta-pinene, cubebol, beta-phellandrene, zingiberene, (*Z*,*Z*)-3,6-nonadienal, and 6-pentyloxan-2-one, which were all derived from Dahongpao gross tea and mainly contributed to the fruity, green, spicy, and woody odor characteristics. Further analysis revealed that beta-myrcene was derived from Dahongpao fresh leaves and its content in TM was significantly higher than that in CK, whereas the six transformed synthetic characteristic volatile compounds were all derived from gross tea, and their contents and odor characteristics were significantly greater in FTM than in FCK. It can be seen that via the first pathway, Dahongpao converted beta-myrcene, which has floral odor characteristics in fresh leaves, into six other characteristic volatile compounds during processing to form the fruity, green, spicy, and woody odor characteristics; this conversion was far more intense in aviation mutagenic Dahongpao than in unmutagenic Dahongpao.

From the second pathway of beta-myrcene conversion, there were five characteristic volatile compounds converted, namely (*E*)-3-hexenyl acetate, (*Z*)-3-hexenyl butyrate, 3-hydroxy-4-methyl-5-ethyl-2-furanone, (*Z*,*Z*)-3,6-nonadienal, and 6-pentyloxan-2-one. Among them, (*E*)-3-hexenyl acetate, (*Z*)-3-hexenyl butyrate, and 3-hydroxy-4-methyl-5-ethyl-2-furanone were converted from Dahongpao fresh leaves, the main odor characteristic of which was fruity, and they were slightly higher in number in CK than in TM. Meanwhile, (*Z*,*Z*)-3,6-nonadienal and 6-pentyloxan-2-one from Dahongpao gross tea, the main odor characteristics of which were fruity and green, were significantly greater in FTM than in FCK. It can be seen that from the second pathway, Dahongpao fresh leaves first converted beta-myrcene with floral odor characteristics into (*E*)-3-hexenyl acetate, (*Z*)-3-hexenyl butyrate, and 3-hydroxy-4-methyl-5-ethyl-2-furanone, which have fruity odor characteristics. Meanwhile, 3-hydroxy-4-methyl-5-ethyl-2-furanone was then converted to (*Z*,*Z*)-3,6-nonadienal and 6-pentyloxan-2-one during processing, which gave the Dahongpao gross tea green fruity odor characteristics, and the intensity of this conversion was significantly higher in aviation mutagenic Dahongpao than in unmutagenic Dahongpao.

From the third pathway of beta-myrcene conversion, there were three characteristic volatile compounds converted, namely 2-p-tolylethanal, 2-methyl-benzaldehyde, and ethyl cinnamate. Among them, 2-p-tolylethanal and 2-methyl-benzaldehyde from Dahongpao fresh leaves, which mainly had a floral odor, were in a significantly higher proportion in TM than in CK, while ethyl cinnamate from the gross tea of Dahongpao, which mainly has a floral odor, was significantly higher in FCK than in FTM. It can be seen that from the third pathway, Dahongpao fresh leaves firstly converted beta-myrcene, which had a floral odor characteristic, into 2-p-tolylethanal and 2-methyl-benzaldehyde, and these two products were then converted to ethyl cinnamate during processing, resulting in Dahongpao gross tea showing stronger floral odor characteristics. The intensity of this conversion was significantly lower in aviation mutagenic Dahongpao than in unmutagenic Dahongpao.

It can be seen that aviation mutagenesis significantly increased the content of beta-myrcene in Dahongpao fresh leaves, and the transformation and synthesis of the compounds during processing were mainly carried out through the first and second pathways of beta-myrcene transformation, which in turn increased the content of beta-pinene, cubebol, beta-phellandrene, zingiberene, (*Z*,*Z*)-3,6-nonadienal, and 6-pentyloxan-2-one and enhanced the intensity of the fruity, green, spicy and woody odor characteristics of Dahongpao gross tea; however, this decreased the intensity of floral odor characteristics. In contrast, beta-myrcene in fresh leaves of unmutagenic Dahongpao was mainly converted and synthesized by the third pathway of compounds during tea processing, which ultimately increased the ethyl cinnamate content in Dahongpao gross tea and enhanced the intensity of floral odor characteristics but decreased the intensity of fruity, green, spicy, and woody notes in Dahongpao gross tea.

## 4. Conclusions

In this study, we analyzed the effects of aviation mutagenesis on volatile compound content and odor characteristics in the fresh leaves and gross tea of Dahongpao, and the results showed that aviation mutagenesis had a small effect on the number and type of volatile compounds but significantly increased the total volatile compound content. Secondly, there were six characteristic volatile compounds that were significantly different for TM compared to CK and seven for FTM compared to FCK. Odor wheel analysis revealed that TM had a significantly more intense floral odor than CK, while its fruity odor was less intense than that of CK. FTM had significantly more intense fruity, green, spicy, and woody odor characteristics than FCK, while its floral odor was significantly less intense. Analysis of the transformation relationship of characteristic volatile compounds revealed that beta-myrcene played an important role in the transformation process of volatile compounds and was a precursor for the transformation and synthesis of other characteristic volatile compounds, and that this compound was mainly derived from fresh leaves of Dahongpao. TM converted beta-myrcene during processing, and beta-pinene, cubebol, beta-phellandrene, zingiberene, (*Z*,*Z*)-3,6-nonadienal, and 6-pentyloxan-2-one contents were mainly enhanced, which in turn improved the fruity, green, spicy, and woody odor characteristics of FTM. On the other hand, CK converted beta-myrcene during processing, and ethyl cinnamate content was mainly increased, which in turn improved floral odor characteristics. This study revealed the effect of aviation mutagenesis on the formation of volatile aroma compounds in leaves of Dahongpao tea tree, and found that aviation mutagenesis can effectively improve leaf quality, which provides a certain reference for the popularization and application of aviation mutagenesis in Dahongpao and in-depth research.

## Figures and Tables

**Figure 1 foods-13-00946-f001:**
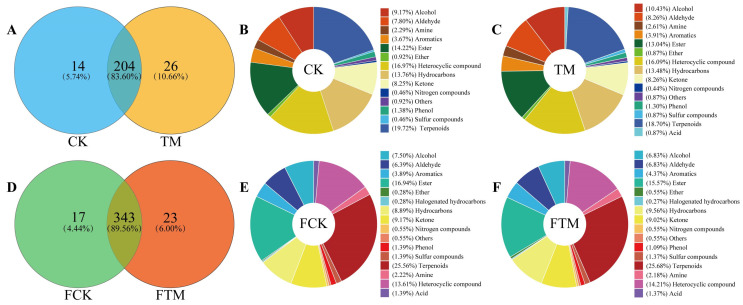
Analysis of the number and classification of volatile compounds in Dahongpao fresh leaves and gross tea. Note: CK: Unmutagenic Dahongpao fresh tea tree leaves; TM: Aviation mutagenic Dahongpao tea tree leaves; FCK: Gross tea processed from fresh leaves of unmutagenic Dahongpao tea tree; FTM: Gross tea processed from fresh leaves of aviation mutagenic Dahongpao tea tree; (**A**): Venn diagram analysis of volatile compounds in CK and TM; (**B**): Classification analysis of volatile compounds in CK; (**C**): Classification analysis of volatile compounds in TM; (**D**): Venn diagram analysis of volatile compounds in FCK and FTM; (**E**): Classification analysis of volatile compounds in FCK; (**F**): Classification analysis of volatile compounds in FTM.

**Figure 2 foods-13-00946-f002:**
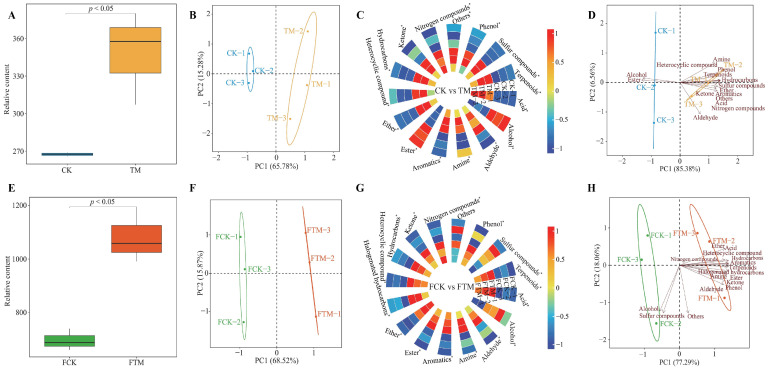
Analysis of the content of volatile compounds in Dahongpao fresh leaves and gross tea. Note: CK: Unmutagenic Dahongpao fresh tea tree leaves; TM: Aviation mutagenic Dahongpao tea tree leaves; FCK: Gross tea processed from fresh leaves of unmutagenic Dahongpao tea tree; FTM: Gross tea processed from fresh leaves of aviation mutagenic Dahongpao tea tree; (**A**): Total amount analysis of volatile compounds in CK and TM; (**B**): PCA analysis of volatile compound content in CK and TM; (**C**): Difference analysis of content after classification of volatile compounds in CK and TM; (**D**): PCA analysis after classification of volatile compounds in CK and TM; (**E**): Total amount analysis of volatile compounds in FCK and FTM; (**F**): PCA analysis of volatile compound content in FCK and FTM; (**G**): Difference analysis of content after classification of volatile compounds in FCK and FTM; (**H**): PCA analysis after classification of volatile compounds in FCK and FTM; * indicates differences between samples at the *p* < 0.05 level.

**Figure 3 foods-13-00946-f003:**
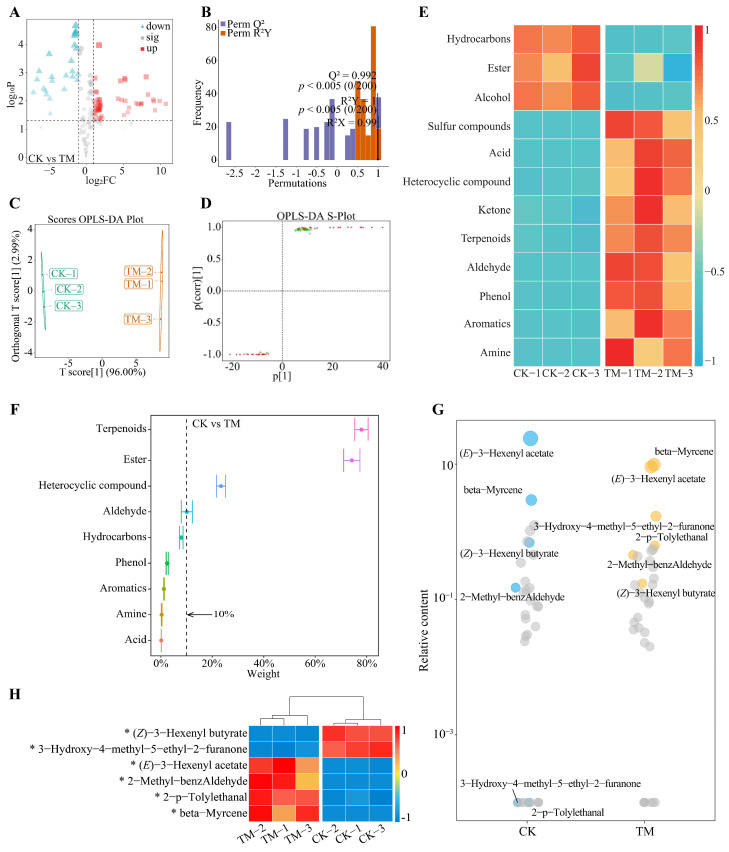
Screening and content analysis of characteristic volatile compounds in fresh leaves of Dahongpao tea tree. Note: CK: Unmutagenic Dahongpao fresh tea tree leaves; TM: Aviation mutagenic Dahongpao tea tree leaves; (**A**): Volcano plot screening for volatile compounds with significant differences in FCK and FTM; (**B**): OPLS-DA model fit test plot for CK and TM; (**C**): OPLS-DA model score plot for CK and TM; (**D**): S-Plot of OPLS-DA model for CK and TM; (**E**): Content analysis after classification of key differential volatile compounds in CK and TM; (**F**): TOPSIS analysis of the impact weights of different classification of key volatile compounds in distinguishing CK from TM; (**G**): Screening for characteristic volatile compounds to distinguish CK from TM by bubble feature plot; (**H**): Heat map analysis of the content of characteristic volatile compounds in CK and TM. * indicates differences between samples at the *p* < 0.05 level.

**Figure 4 foods-13-00946-f004:**
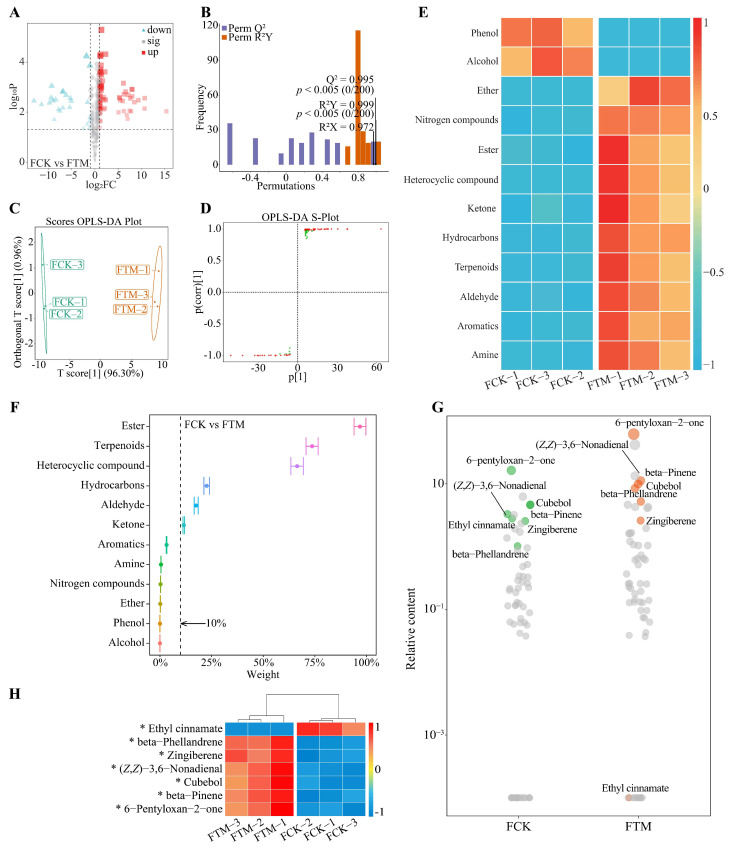
Screening and content analysis of characteristic volatile compounds in Dahongpao gross tea. Note: FCK: Gross tea processed from fresh leaves of unmutagenic Dahongpao tea tree; FTM: Gross tea processed from fresh leaves of aviation mutagenic Dahongpao tea tree; (**A**): Volcano plot screening for volatile compounds with significant differences in FCK and FTM; (**B**): OPLS-DA model fit test plot for FCK and FTM; (**C**): OPLS-DA model score plot for FCK and FTM; (**D**): S-Plot of OPLS-DA model for FCK and FTM; (**E**): Content analysis after classification of key differential volatile compounds in FCK and FTM; (**F**): TOPSIS analysis of the impact weights of different classification of key volatile compounds in distinguishing FCK from FTM; (**G**): Screening for characteristic volatile compounds to distinguish FCK from FTM by bubble feature plot; (**H**): Heat map analysis of the content of characteristic volatile compounds in FCK and FTM. * indicates differences between samples at the *p* < 0.05 level.

**Figure 5 foods-13-00946-f005:**
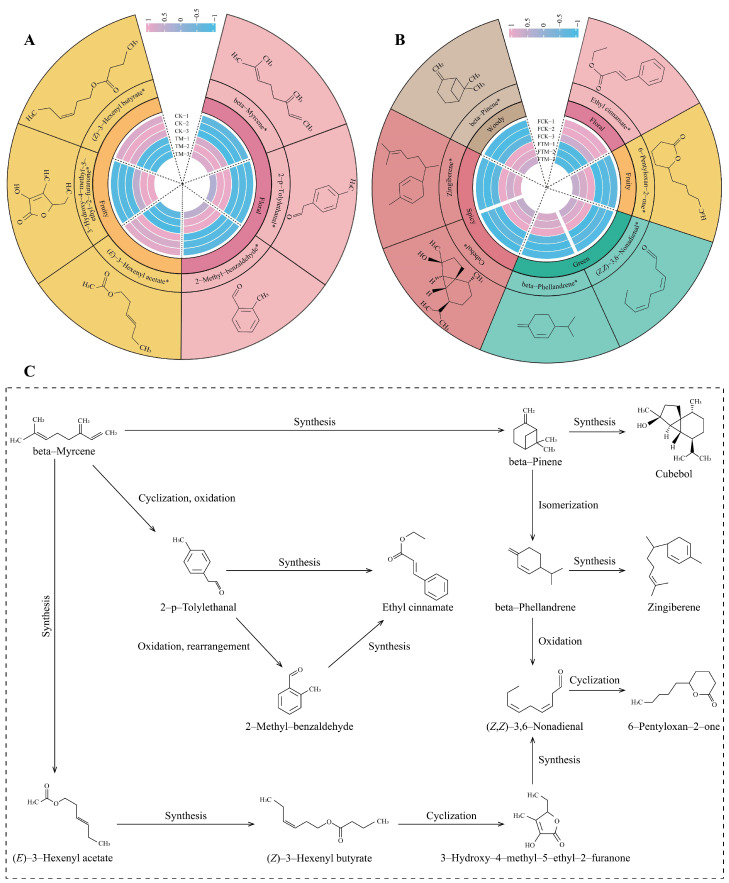
Odor and transformation pathway analysis of characteristic volatile compounds in Dahongpao fresh leaves and gross tea. Note: CK: Unmutagenic Dahongpao fresh tea tree leaves; TM: Aviation mutagenic Dahongpao tea tree leaves; FCK: Gross tea processed from fresh leaves of unmutagenic Dahongpao tea tree; FTM: Gross tea processed from fresh leaves of aviation mutagenic Dahongpao tea tree; (**A**): Odor characteristics and their intensity analysis of characteristic volatile compounds in CK and TM; (**B**): Odor characteristics and their intensity analysis of characteristic volatile compounds in FCK and FTM; (**C**): Transformation pathway analysis of characteristic volatile compounds. * indicates that the difference between different samples reaches the *p* < 0.05 level.

## Data Availability

The data presented in this study are available on request from the corresponding author. The data are not publicly available due to privacy restrictions.

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
