# Peer review of "Aviation Mutagenesis Alters the Content of Volatile Compounds in Dahongpao (Camellia sinensis) Leaves and Improves Tea Quality"

_foods, 2024, doi:10.3390/foods13060946_

Round 1

Reviewer 1 Report

Comments and Suggestions for Authors

I believe that the manuscript “Aviation mutagenesis alters the content of volatile compounds in Dahongpao (Camellia sinensis) leaves and improves the aroma quality of tea” may be of interest to readers of the journal.

Dear authors, I enclose some suggestions for improving the article

Title: In the title the authors mention the aroma quality of tea, however, the manuscript does not contain a qualitative evaluation of the aroma which must be carried out through sensory analysis. Therefore, authors need to remove the aroma word from the title.
Figure 3 is too crowded, better to divide it into two figures. Furthermore, G (screening of characteristic volatile compounds to distinguish CK from TM by bubble feature plot) must have better image quality.
Figure 4: idem
Conclusions: In their conclusions, the authors must rule out the possibility that aerospace mutagenesis improves the aroma quality of tea, as their study only showed an effect on volatile compounds and not on aroma quality.

Author Response

Comments and Suggestions for Authors

I believe that the manuscript “Aviation mutagenesis alters the content of volatile compounds in Dahongpao (Camellia sinensis) leaves and improves the aroma quality of tea” may be of interest to readers of the journal.

Dear authors, I enclose some suggestions for improving the article

Title: In the title the authors mention the aroma quality of tea, however, the manuscript does not contain a qualitative evaluation of the aroma which must be carried out through sensory analysis. Therefore, authors need to remove the aroma word from the title.

A: Thank you to the reviewer. The authors have removed it.

Figure 3 is too crowded, better to divide it into two figures. Furthermore, G (screening of characteristic volatile compounds to distinguish CK from TM by bubble feature plot) must have better image quality.

A: Thank you to the reviewer. Because this figure is a whole, it is better not to divide it into two figures. To avoid crowding, the authors have adjusted the graph while improving the quality of Figure 3G.

Figure 4: idem

A: Thank you to the reviewer. The authors have made adjustments based on the experts' comments.

Conclusions: In their conclusions, the authors must rule out the possibility that aerospace mutagenesis improves the aroma quality of tea, as their study only showed an effect on volatile compounds and not on aroma quality.

A: Thank you to the reviewer. The authors have made changes in the conclusion.

Reviewer 2 Report

Comments and Suggestions for Authors

The manuscript presents an exterelmy interesting approach to obtaine new, possibly improved products. Authors for sure had an unique opportuinity to perfome the aviation muthagenesis of C. sinensis.

The manuscript presents an consist idea, which will for sure find a lot of public attention. The big strenght of this manuscript armodern statistical tools applied for the analysis and the way of results presentation. 

Non the less in analytical part the manuscript requires a great improvments. The method for volatiles analysis are not clear, especially in terms of identification method of analytes and quantification method.

lso, the list of volatiles identified (presented in supplementary files) is strongly odd. In my opinion the identification process was done by automatic algorithm, which usally presents a number of errors. Also, it is uncommon to present the systematic volatile compound names. For instance, (E)-1-Methyl-4-(6-methylhept-5-en-2-ylidene)cyclohex-1-ene given in the table, normally is presented as (E)-γ-Bisabolene

 Since the volatile identification and quantification is the  core of this work, I have strong doubts if the data and conclusions based on them are reliable.

At this moment I will advise to reconsider the manuscript after major revision, however Authors have a great work to do to prove that the manuscript is reliable. The crucial part is to clarify the qualitative and quantitative analysis issue and to in-depth verify and rebulit the tables with identified voltiles.

My full review is given as pdf file.

Comments on the Quality of English Language

Minor editing of English language required

Author Response

Comments and Suggestions for Authors

The manuscript presents an exterelmy interesting approach to obtaine new, possibly improved products. Authors for sure had an unique opportuinity to perfome the aviation muthagenesis of C. sinensis.

The manuscript presents an consist idea, which will for sure find a lot of public attention. The big strenght of this manuscript armodern statistical tools applied for the analysis and the way of results presentation. 

Non the less in analytical part the manuscript requires a great improvments. The method for volatiles analysis are not clear, especially in terms of identification method of analytes and quantification method.

A: Thank you to the reviewer. The authors have added qualitative and quantitative methods in Materials and Methods.

lso, the list of volatiles identified (presented in supplementary files) is strongly odd. In my opinion the identification process was done by automatic algorithm, which usally presents a number of errors. Also, it is uncommon to present the systematic volatile compound names. For instance, (E)-1-Methyl-4-(6-methylhept-5-en-2-ylidene)cyclohex-1-ene given in the table, normally is presented as (E)-γ-Bisabolene

 Since the volatile identification and quantification is the core of this work, I have strong doubts if the data and conclusions based on them are reliable.

A: Thank you to the reviewer. "(E)-1-Methyl-4-(6-methylhept-5-en-2-ylidene)cyclohex-1-ene" is named according to the "Nomenclature of Organic Chemistry" published by the International Union of Pure and Applied Chemistry (IUPAC) and "Principles of Organic Chemistry Nomenclature" published by the Chinese Chemical Society, and "(E)-γ-Bisabolene" is the English alias of the compound. Compounds may have multiple English aliases, but their standard nomenclature is unique. For each compound, the authors have provided the compound's CAS number, by which the compound and all of its English aliases can be looked up. Therefore, the compounds provided by the authors in the Supplementary Data are correct and the data are reliable. Once again, we thank the reviewers for their suggestions.

At this moment I will advise to reconsider the manuscript after major revision, however Authors have a great work to do to prove that the manuscript is reliable. The crucial part is to clarify the qualitative and quantitative analysis issue and to in-depth verify and rebulit the tables with identified voltiles.

A: Thank you to the reviewer. The authors have added qualitative and quantitative methods in Materials and Methods. The authors also state that the compounds in the Supplementary Data are correct, and the authors use standard compound nomenclature for naming the compounds, as well as providing the CAS number for each compound. Many thanks to the reviewers for their careful review.

My full review is given as pdf file.

1.“Z”italic

A: Thank you to the reviewers. The authors have revised the full manuscript.

2. I advise to not use 'tea tree' term. it may be confused with other plant - Melaleuca alternifolia, which commonly is named tea tree.

A: Thank you to the reviewers. The authors have revised the name to "Camellia sinensis" and added "tea tree (Camellia sinensis)" to the first occurrence of "tea tree".

3. An Introduction requires a paragraph which will present the up-to-date knowledge regarding Camellia sinensis volatile compounds.

A: Thank you to the reviewers. The authors have made appropriate additions.

4. Please confirm the linear velocity value. Normally linera velocity is expressed as cm3/sec, while column flow is expressed as mL/min. The column flow should be around 1 mL/min, and the linear velocity 30-40 cm3/sec. Of course the given value after calculation may be correct, however it should be checked. If possible present linear velocity as cm3/sec.

A: Many thanks to the reviewers. Both representations are feasible. The "1.2 mL/min" in the manuscript is correct. Thank you very much for your careful review.

5. Below you write that the detector temperature was 150. Please check.

A: Thanks to the reviewer . The authors have clarified in Materials and Methods that 150°C is the temperature of the quadrupole, not the detector temperature, which is 280°C.

6. Please give more details regarding identification and quantification. What techniques were used for identification? Which databases. Moreover,looking on the list of identified compounds it was not a target analysis. In that case SIM mode would not be possible to use, since it will not give enough data for databases. What quantifcation method was? used? in supplementary data there is no unit.

A: Thanks to the reviewers. The authors have added quantitative and qualitative methods in the Materials and Methods. The database is the NIST20 Mass Spectrometry Database. Also, in the Supplementary Material, the authors have added the quantitative, quantitative ion and NIST20 assays for each compound. Secondly, the quantitative analysis uses relative quantification, its unit is "mAU*min". The authors have also added this in the supplementary material.

7. “Glycyrrhiza glabra” italic

A: Thank you to the reviewers. The authors have revised it.

8. The quality of Fig.5. A and B is not too good. Is it possible toimprove? I think authors should work with Shem presented in Fig.5C.  The changes are not logical.  For instance:  2-p-Tolyethanal --> 2-Methyl-benzaldehyde: Authors says that this step requires oxidation and rearengment, while the first compound has 9 carbons and second compound 8  carbons, so something else has to be done.   (Z,Z)-3,6-Nonadienal --> 6-Pentylolaxan-2-one: It is not only cyclization, but also additional oxigen is introduced.

A: Thank you to the reviewer. Figure5A and figure 5B may be blurred after being converted to PDF. The author has replaced images with ones of greater clarity. Second, for Figure 5C, the authors listed only the most important transformations, and the other details in the transformation process are described in more detail by the authors in the results and analysis. Many thanks to the reviewers for their careful review.

9. Keep with Z and E nomenclature.

A: Thank you to the reviewers. The authors have revised the full manuscript.

Reviewer 3 Report

Comments and Suggestions for Authors

Dear Authors,

In general, the review is good to read, the structure of the work is clear. Generally, after reading, the review seems almost complete. Nevertheless, I have a few comments that will improve the substantive quality of the work. See below.

1.      General note for the future. For convenience to do a review, please mark the lines with numbers throughout the document.

2.      The great disadvantage of the work is that no field tests were carried out in a two- or three-year cycle. The tea was harvested only in the 2023 season. Please comment it. This creates uncertainty. Are the differences in VOCs due 100% to aviation mutagenesis or to different growing conditions. For example, other exposure to light, wind, precipitation etc...

3.      Introduction. Wuyi Mountain, Fujian, China (27°32′36″27°55′15″ N, 117°24′12″118°02′50″ E) is an important tea-producing area in China, the birthplace of oolong tea. Wuyi Mountain has uniform temperature in all seasons, mild and humid, and its geology belongs to the typical Danxia landform, the special geographic environment creates a special type of tea - Wuyi Rock Tea". What is this average temperature?

4.      "2.2. Volatile metabolome analysis" - did the authors use any libraries to identify compounds? If so, please add it.

5.      Chapter: "3.1. Analysis of the number and classification of volatile compounds in Dahongpao fresh leaves and gross tea".

6.      Text from "Aerospace mutagenesis is one of........." to "on volatile compounds in Dahongpao fresh leaves and gross tea". It should rather be in Introduction or Materials and Methods.

7.      Conclusion chapter. According to the authors, which factor may have the greatest impact on changes in VOCs, changes in the magnetic field, gravity or radiation in space? Such a comment would be valuable.

Author Response

Comments and Suggestions for Authors

Dear Authors,

In general, the review is good to read, the structure of the work is clear. Generally, after reading, the review seems almost complete. Nevertheless, I have a few comments that will improve the substantive quality of the work. See below.

  1. General note for the future. For convenience to do a review, please mark the lines with numbers throughout the document.

A: Many thanks to the reviewer. After the manuscript was submitted, the editing system automatically converted the formatting, resulting in some missing line numbers in the reviewer manuscript. I am very sorry for the inconvenience caused to your review.

  1. The great disadvantage of the work is that no field tests were carried out in a two- or three-year cycle. The tea was harvested only in the 2023 season. Please comment it. This creates uncertainty. Are the differences in VOCs due 100% to aviation mutagenesis or to different growing conditions. For example, other exposure to light, wind, precipitation etc...

A: Many thanks to the reviewer. Your suggestions are very good. In 2011, tea tree seeds were sent to space for aerial mutagenesis. After returning to the ground and planting, it also takes 7 years to harvest according to the normal growth of the tea tree. Given that the tea tree seeds in question were the first to be sent into space by China, and that they are scarce, they are more valuable. The local government protects them and is unable to conduct scientific research. In recent years, it was only through communication and the signing of cooperation agreements that the group was able to formalize this aspect of the research. Secondly, CK and TM were planted in the same plot, and the authors provided photographs of where they were planted in the supplementary material, where the environmental conditions of light, wind and precipitation were the same. Therefore, the main reason for the difference in volatile compounds is due to aerial mutagenesis. Many thanks to the reviewers.

  1. Introduction. Wuyi Mountain, Fujian, China (27°32′36″~27°55′15″ N, 117°24′12″~118°02′50″ E) is an important tea-producing area in China, the birthplace of oolong tea. Wuyi Mountain has uniform temperature in all seasons, mild and humid, and its geology belongs to the typical Danxia landform, the special geographic environment creates a special type of tea - Wuyi Rock Tea". What is this average temperature?

A: Thank you to the reviewer. The authors have added.

  1. "2.2. Volatile metabolome analysis" - did the authors use any libraries to identify compounds? If so, please add it.

A: Thank you very much to the reviewer. The database used by the authors is "NIST20 Mass Spectrometry Database". The authors have supplemented it in Materials and Methods.

  1. Chapter: "3.1. Analysis of the number and classification of volatile compounds in Dahongpao fresh leaves and gross tea".

A: Thank you to the reviewers. The authors have revised it.

  1. Text from "Aerospace mutagenesis is one of........." to "on volatile compounds in Dahongpao fresh leaves and gross tea". It should rather be in Introduction or Materials and Methods.

A: Thank you to the reviewers. The authors have adjusted it to the introduction.

  1. Conclusion chapter.According to the authors, which factor may have the greatest impact on changes in VOCs, changes in the magnetic field, gravity or radiation in space? Such a comment would be valuable.

A: Many thanks to the reviewers. It is a good idea, but due to the limitation of experimental conditions, the authors were unable to simulate the space conditions for a one-factor experimental study. Therefore, the authors are unable to specify which of the factors of magnetic fields, gravity, and space radiation plays the most important role. It can only be stated that the aviation mutagenesis is the result of a combination of factors. Many thanks to the reviewers.